# The Degree of Problematic Technology Use Negatively Affects Physical Activity Level, Adherence to Mediterranean Diet and Psychological State of Adolescents

**DOI:** 10.3390/healthcare11121706

**Published:** 2023-06-11

**Authors:** Adrián Mateo-Orcajada, Raquel Vaquero-Cristóbal, Mario Demófilo Albaladejo-Saura, Lucía Abenza-Cano

**Affiliations:** Facultad de Deporte, UCAM Universidad Católica de Murcia, 30107 Murcia, Spain; amateo5@ucam.edu (A.M.-O.); mdalbaladejosaura@ucam.edu (M.D.A.-S.); labenza@ucam.edu (L.A.-C.)

**Keywords:** basic psychological needs, life satisfaction, new technologies, physical activity, physical fitness, youth

## Abstract

The previous scientific literature has shown how detrimental addictive internet and mobile phone use can be for the adolescent population. However, little is known about their influence on the physical activity, kinanthropometry and body composition, nutrition patterns, psychological state, and physical fitness of this population. For this reason, the objectives of this research were (a) to determine the differences in the physical activity level, kinanthropometric and body composition variables, adherence to Mediterranean diet (AMD), psychological state, and physical fitness according to gender and different levels of problematic use of the internet and mobile phones; and (b) to establish the differences in the physical activity level, kinanthropometric and body composition variables, AMD, psychological state, and physical fitness among adolescents when considering problematic use of the internet and mobile phones in combination. The sample consisted of 791 adolescent males and females between 12 and 16 years of age (1st to 4th course) from four compulsory secondary schools (404 males and 387 females; mean age: 14.39 ± 1.26 years-old; mean height: 163.47 ± 8.94 cm; mean body mass: 57.32 ± 13.35 kg; mean BMI: 21.36 ± 3.96 kg/m^2^). The physical activity level (baseline score: 2.64 ± 0.67), kinanthropometric variables and body composition, AMD (baseline score: 6.48 ± 2.48), psychological state (baseline life satisfaction: 17.73 ± 4.83; competence: 26.48 ± 7.54; autonomy: 25.37 ± 6.73; relatedness: 24.45 ± 6.54), and physical condition variables were measured. The results showed that adolescent males and females with problematic internet and/or mobile phone use presented a worse psychological state, but it is especially relevant to highlight that females also had a lower level of physical activity and AMD, with problematic mobile phone use being especially relevant, mainly in the psychological state of adolescents. In conclusion, problematic use of the internet and mobile phones can have detrimental effects on the level of physical activity, AMD, and psychological state of adolescents, with the differences found in females being particularly relevant.

## 1. Introduction

The COVID-19 pandemic exacerbated the excessive use of mobile phones in the adolescent population [1]. Scientific research conducted during and after the pandemic found that a percentage of nearly 90% of adolescents used mobile phones on a regular basis. Along with the increase in the use of these devices, problematic use of the internet and mobile phones also increased, understood as a pattern of interaction with the mobile phone and the internet that is characterized by repetitive use of the mobile phone and internet to engage in negative health behaviors [2,3]. This includes the inability to regulate mobile phone and internet use, leading to associated negative consequences in daily life, including technological dependence, as well as social, behavioral, and affective problems [2,3]. This problematic use reaches rates close to 35% of adolescents [4]. In addition, it should be noted that this problematic use of mobile phones and internet does not affect adolescent males and females equally, and significant differences have been found depending on gender [5,6,7].

The problematic use of these technologies affects the possibilities of the healthy development of adolescents [8], with the excessive use of the internet on weekdays, excessive time spent playing online games, and not being able to use social networks being factors that exert a negative impact on this population [8]. Thus, there are numerous adolescents who use the internet and mobile phones daily, and they do so in a problematic way, characterized by excessive or maladaptive use that disrupts the individual’s social functions and involves withdrawal, compulsive behavior, and functional impairment [9], which could be affecting their health status [10].

Numerous aspects of adolescent health are affected by problematic internet and mobile phone use, with a special emphasis on physical activity practice [6,11], body composition and kinanthropometric variables [12,13], nutritional habits [6,14], psychological state [11,15], and physical fitness [16,17].

Regarding the practice of physical activity, previous results are contradictory, as some research showed that adolescents with problematic internet use performed a significantly lower level of sports practice [6,11], while other research found no significant differences between adolescents with and without problematic use [18]. According to kinanthropometric and derived variables, a higher body mass index and a higher probability of being overweight were found in adolescents who used their mobile phones for more than three hours a day [12,13]. However, no study has attempted to investigate differences in variables related to adiposity, muscle mass development, or body composition in adolescents who present problematic internet and mobile use. In addition, no previous research has analyzed whether internet use, mobile use, or both in combination are equally detrimental to physical activity level and body composition in this population.

Similarly, young people with problematic internet use showed irregular dietary behaviors characterized by a reduced consumption of fruits and vegetables, skipping meals, snacking abuse, consumption of carbonated soft drinks, and consumption of fast food, resulting in a poor-quality diet [6,14]. However, no previous research has analyzed the differences in a nutritional dietary pattern, which is much more representative of the daily diet, such as adherence to the Mediterranean diet (AMD), according to the level of problematic use of the internet and mobile phones. This dietary pattern has been widely used in the adolescent population because it has been shown to be one of the best nutritional structured recommendations and includes numerous health-related nutritional lifestyle habits [19,20]. In addition, previous research has shown that this index is related to other healthy and fundamental variables for the development of the adolescent population, such as body composition [21], physical condition [22], and level of physical activity [23]. Therefore, it is necessary to analyze the differences in adherence to this nutritional pattern according to the level of problematic internet and mobile phone use, as it is unknown in the previous scientific literature and may be an aspect of great relevance for adolescent development.

Regarding psychological state, numerous previous studies have analyzed its relationship with the problematic use of technology, obtaining similar conclusions in which they highlighted that adolescents who showed problematic use of these technologies saw their psychological state negatively affected, increasing the likelihood of suffering depression and anxiety [11,15]. However, previous research has not analyzed existing differences in the satisfaction of basic psychological needs and life satisfaction of adolescents according to their level of problematic internet and mobile phone use. This is especially relevant given the relationship between these psychological variables and healthy behaviors such as nutrition or physical activity [24,25]. In addition, it is not known whether problematic use of the internet and mobile phones together is more detrimental to the psychological state of adolescents than problematic use of one of these technologies.

With respect to physical fitness, adolescents with problematic use of mobile phones showed reduced physical fitness, which considerably affected their health [17]. Little is known about the relationship between internet and mobile phone use and differences in the physical fitness of adolescents, as only one study, by Bravo-Sánchez et al. [16], provided specific results on physical fitness tests (abdominal test, medicine ball throw, broad jump, 50 m sprint, deep trunk flexion, and agility test). However, this research only recorded the level of mobile phone use, so it is unknown whether internet use is also detrimental to adolescents’ physical fitness, and whether the two combined (mobile and internet) may have even more detrimental effects.

Based on previous scientific research, there is a gap in the previous scientific literature in which it is unknown whether the degree of problematic internet or mobile phone use by adolescents is related to a worse level of physical activity, kinanthropometric and body composition variables, AMD, psychological state, and physical fitness. Lastly, to all of the above, it should be added that these previous studies did not analyze the influence of gender in the relationship between the problematic use of technology and the variables mentioned above, and that no previous research has analyzed the combination of problematic internet and mobile phone use in the adolescent population; therefore, it is unknown whether one or the other is more relevant in the existence of differences in the level of physical activity, kinanthropometric and body composition variables, AMD, psychological state, and physical fitness.

For this reason, the aims of the present investigation were (a) to determine the differences in the physical activity level, kinanthropometric and body composition variables, AMD, psychological state, and physical fitness according to gender and different levels of problematic use of the internet and mobile phones; and (b) to establish the differences in the physical activity level, kinanthropometric and body composition variables, AMD, psychological state, and physical fitness among adolescents when considering problematic use of the internet and mobile phone in combination.

## 2. Materials and Methods

### 2.1. Design

A cross-sectional design was followed, in accordance with the STROBE guidelines [26]. A non-probabilistic convenience sampling was used, selecting those adolescents to whom access was available from the educational centers contacted. The Institutional Ethics Committee of the Catholic University of Murcia reviewed and authorized the protocol designed for data collection in accordance with the World Medical Association (code: CE022102). The Declaration of Helsinki statements were followed throughout the entire process.

All measurements took place on the same day, utilizing the dedicated time slot of the physical education class and the enclosed sports pavilion at the education centers as the designated location. This approach aimed to minimize the presence of confounding variables that could potentially affect the results.

### 2.2. Participants

A total of 791 adolescents aged 12 to 16 years old participated in the study (404 males; 387 females; mean age: 14.39 ± 1.26 years-old). The adolescents belonged to four compulsory secondary schools in different areas of the Region of Murcia, all of them public schools located in developed areas. The level of sports practice was low (mean physical activity level: 2.64 ± 0.67). The students belonged to the four academic years of compulsory secondary education (1st to 4th course), distributed as follows: 1st (*n* = 206), 2nd (*n* = 133), 3rd (*n* = 238), and 4th (*n* = 214). The inclusion criteria were a) attending compulsory secondary education; b) not presenting any disability that prevented the completion of the tests; and c) age between 12 and 16 years old; participants were excluded in case of a) a change of school during the academic year; and b) not completing the questionnaires, kinanthropometric measurements, or physical fitness tests in their entirety.

The sample selection flowchart is shown in Figure 1. The sample size was calculated using the Rstudio statistical software (v. 3.15.0; Rstudio Inc., Boston, MA, USA) and setting the statistical significance at α = 0.05. Standard deviations (SD) from previous research that had analyzed problematic internet (SD = 4.50) and mobile phone use in young people (SD = 3.50) [27] were used. This technique for sample size calculation is based on the use of the SD, a constant, and the estimated effect size, and has been used in previous research [28]. With an estimated error (d) of 0.32 for the internet score and 0.25 for the mobile phone score, for a 95% confidence interval (CI), the minimum sample needed was 756 adolescents.

A non-probabilistic convenience sampling was carried out by contacting the schools in the different areas of the Region of Murcia (north, west, east, and south) that had the largest number of students in compulsory secondary education. The four high schools contacted decided to participate voluntarily in the study. The adolescents who were willing to participate voluntarily completed the informed consent form, after which it was signed by them and their parents.

### 2.3. Instruments

#### 2.3.1. Questionnaires

The Questionnaire of Experiences Related to Internet use (CERI) and the Questionnaire of Experiences Related to Mobile phones (CERM) were used to analyze the adolescents’ problematic use of the internet and mobile phones [29]. Both questionnaires had been previously validated, and had a Cronbach’s alpha of 0.77 for the CERI and 0.80 for the CERM [29]. Both instruments are composed of 10 items that are completed with a Likert scale of 1 to 4 points (1: never; 4: almost always), with the sum of the 10 items being the final score of the questionnaire. Both questionnaires measure problematic use (PU), when the score is higher than 26 in CERI and 24 in CERM; occasional problems (OP), when the score is between 18 and 25 in CERI and 16 and 23 in CERM; and the absence of problems (NP), when the score is lower than 18 in CERI and 16 in CERM [29].

The level of physical activity performed was measured using the Physical Activity Questionnaire for Adolescents (PAQ-A), an instrument that was previously validated, and which obtained an intraclass correlation coefficient of 0.71 [30]. This questionnaire is composed of nine items that measure physical activity in the previous week using a Likert scale of 1 to 5 points (1: no physical activity; 5: a lot of physical activity). The final score is obtained from the arithmetic mean of the scores from the first eight items, with 1 being the minimum score and 5 the maximum.

The “Mediterranean Diet Quality Index for Children and Adolescents” (KIDMED) was used to assess the adherence of adolescents to the Mediterranean nutritional pattern, providing the final AMD score [19]. This questionnaire has a good agreement in its general score (kappa = 0.73) for use in the adolescent population [31]. To complete the 16 items that shaped the questionnaire, adolescents had to indicate whether they complied with the statement indicated. The score of the responses varied between +1 (positive connotation, 12 items) and −1 (negative connotation, 4 items), resulting in a final score between 0 and 12 points [19].

The psychological state of the adolescents was assessed by means of the Satisfaction with Life Scale (SWLS) [32] and the Basic Psychological Needs Scale (BPNS) [33]. The SWLS is composed of 5 items that are answered with a Likert scale from 1 to 5, where the final score is obtained by adding all the items, with a minimum score of 5 and a maximum of 25; the BPNS is composed of 18 items (6 per dimension: competence, autonomy, and relatedness) completed with a Likert scale from 1 to 6 points, where the final score is obtained by adding all the items, with a minimum score of 6 and a maximum of 36. Both scales have adequate external validity and internal consistency for use with adolescents (life satisfaction: α = 0.84; competence: α = 0.80; autonomy: α = 0.69; and relatedness: α = 0.73) [34,35].

#### 2.3.2. Kinanthropometric and Body Composition Measurement

The kinanthropometric measurements were carried out by three accredited anthropometrists from the International Society for the Advancement of Kinanthropometry (ISAK), ranging from levels 2 to 4. The measurements encompassed two fundamental variables: body mass and height. Body mass was determined using a TANITA BC-418-MA Segmental scale (TANITA, Tokyo, Japan) with a precision of 100 g, while height was measured using a SECA stadiometer 213 (SECA, Hamburg, Germany) with a precision of 0.1 cm. Additionally, three skinfold measurements were taken on the triceps, thigh, and calf using a skinfold caliper (Harpenden, Burgess Hill, UK) with an accuracy of 0.2 mm. Furthermore, five girth measurements were recorded, including relaxed arm, waist, hips, thigh, and calf, employing an inextensible Lufkin W606PM tape (Lufkin, Missouri City, TX, USA) with a precision of 0.1 cm. Before conducting the measurements, all instruments were properly calibrated, and the entire process adhered to the standardized protocol established by the ISAK [36].

It is important to highlight that each subject’s measurements were conducted consistently by the same anthropometrist. A minimum of two measurements were obtained for each variable, and a third measurement was taken if the difference between the initial two measurements exceeded 5% for the skinfolds or 1% for the remaining measurements. The final value utilized for the analysis was determined based on the mean of the measurements when two were taken, or the median when three measurements were recorded. This approach ensured reliable and accurate data for each variable [36].

The intra- and inter-evaluator technical error of measurements (TEM) were calculated for a subsample. These were 0.02% and 0.03% for the basic measurements, 1.21% and 1.98% for the skinfolds, and 0.04% and 0.06% for the girths, respectively, and their correlation coefficients with an expert, level 4 anthropometrist were 0.96 for the basic measurements, 0.84 for the skinfolds, and 0.87 for the girths.

In addition, the following variables, derived from the kinanthropometric measurements, were calculated: BMI, fat mass (%) [37], muscle mass [38], Σ3 skinfolds (triceps, thigh, and calf), waist-to-hip ratio (waist girth/hip girth), and corrected girths (arm, thigh, and calf). Corrected girths were calculated using the following formulas: arm [arm relaxed girth-(π × triceps skinfold)], thigh [thigh girth-(π × thigh skinfold)], and calf [calf girth-(π × calf skinfold)].

#### 2.3.3. Physical Fitness Measurement

According to the methodology of previous research, the following physical fitness tests were carried out [39].

The 20 m shuttle run test, an incremental test with high validity and reliability for use with adolescents, was used to assess the cardiorespiratory capacity of the adolescents [40]. Using the formula by Léger et al. (1988) and the speed at which the subject left the test, the maximum oxygen consumption (VO2 max.) was predicted.

Upper and lower limb strength were assessed using the handgrip strength test [41] and the countermovement jump test (CMJ) [42], respectively. To assess upper limb strength, participants were instructed to utilize a Takei Tkk5401 digital handheld dynamometer (Takei Scientific Instruments, Tokyo, Japan) and squeeze it with their elbow fully extended. This position was chosen, as it allows for maximum force production [43]. For the CMJ, a force platform with a sampling frequency of 200 Hz (MuscleLab, Stathelle, Norway) was employed. During the CMJ test, participants were directed to execute a maximal vertical jump while keeping their hands on their waists and ensuring full extension of the trunk throughout the flight phase [42].

To measure hamstring and lower back flexibility, the sit-and-reach test was used [44]. Following the protocol from previous research [45], adolescents had to perform a maximum trunk flexion, keeping their hands and knees fully extended, to reach the maximum distance possible by sliding the palms of their hands, one on top of the other, across the box.

To measure speed, the 20 m sprint was used, in which the adolescents, starting from a standing position behind a line, and initiating the race at a time of their choice, had to cover 20 m in the shortest possible time [46]. For the measurement, single-beamed photocells (Polifemo Light; Microgate, Italy) placed at hip height were used [47,48].

For each of the physical tests, except for the sit-and-reach and the 20 m shuttle run test, two attempts were made, leaving two minutes of recovery time between each attempt, and five minutes between each physical condition test. The best value obtained in the two attempts was considered for analysis. The physical fitness tests were overseen by four researchers with experience in the field.

### 2.4. Procedure

First, the participants completed the CERI, CERM, PAQ-A, KIDMED, SWLS, and BPNS questionnaires. Second, the kinanthropometric measurements were taken. Third, the sit-and-reach test was performed, prior to the warm-up, because this could influence test performance [49]. Fourth, the handgrip strength, CMJ, and 20 m sprint tests were explained to the adolescents, and once they were familiar with the protocol, a general warm-up was performed that included 5 min of progressive running and 10 min of mobility of the main joints involved in the physical fitness tests. A researcher oversaw the warm-up and randomly indicated the physical test to be performed to each adolescent. Fifth, and once the handgrip strength, CMJ, and 20 m sprint tests had been completed, the 20 m shuttle run test was performed. The order of the physical fitness tests was chosen according to the recommendations of the fatigue generated and metabolic demand for each test established by the National Strength and Conditioning Association (NSCA) [50].

### 2.5. Data Analysis

The normality of the data was assessed using the Kolmogorov–Smirnov test, skewness, and kurtosis, and since all the variables had a normal distribution, parametric tests were used for their analysis. Descriptive statistics were used to determine the mean and standard deviation. Two one-factor ANOVAs were performed: the first to analyze the differences between males with different degrees of problematic use, and females with different problematic internet and mobile phone use in the study variables; and the second to establish the differences between adolescents when considering the combination of the different degrees of problematic internet and mobile phone use. To determine the significant differences between groups for each variable, Bonferroni’s pairwise comparison was employed. The effect size (ES) was calculated using partial eta squared (η^2^), where ES values were categorized as small (ES ≥ 0.10), moderate (ES ≥ 0.30), large (ES ≥ 1.2), or very large (ES ≥ 2.0), with the significance level set at *p* < 0.05 [51]. Statistical significance for the conducted tests was defined as *p* < 0.05. The statistical analysis was carried out using the SPSS statistical package (v.25.0; SPSS Inc., Chicago, IL, USA).

## 3. Results

The differences between males with different levels of problematic internet and mobile phone use and females with different levels of problematic internet and mobile phone use are shown in Table 1. The results show significant differences with problematic internet use in the level of physical activity (*p* = 0.013), AMD (*p* = 0.001), life satisfaction (*p* = 0.009) and competence (*p* = 0.009) in females; meanwhile, in males, the differences were found in right arm handgrip (*p* = 0.001), left arm handgrip (*p* = 0.019), and CMJ (*p* = 0.014). Regarding problematic mobile phone use, differences were significant in AMD (*p* < 0.001–0.047) and life satisfaction (*p* < 0.001–0.001) in both males and females, as well as in relatedness (*p* = 0.010) and CMJ (*p* = 0.003) in males.

With respect to internet use, PU females showed a worse physical activity score (*p* = 0.012), AMD (*p* = 0.001), life satisfaction (*p* = 0.009), and competence (*p* = 0.023) than NPs, as well as a worse score than OPs in physical activity score (*p* = 0.031) and AMD (*p* = 0.028). Regarding males, NPs showed higher scores in life satisfaction (*p* = 0.048) than OPs, while PUs showed higher scores than OPs and NPs in right arm handgrip (*p* < 0.001–0.018), left arm handgrip (*p* = 0.018–0.048), and CMJ (*p* = 0.015–0.028) (Figure 2).

Regarding mobile phone use (Figure 2), NP females showed higher scores than OPs and PUs in AMD (<0.001) and life satisfaction (*p* = 0.003). For males, OPs showed a lower score than NPs in life satisfaction (*p* = 0.001), but a higher one than PUs in relatedness (*p* = 0.010); meanwhile, NPs showed a lower score than OPs (*p* = 0.018) and PUs (*p* = 0.025) in CMJ.

Table 2 shows the differences when considering the problematic use of the internet and mobile phone in combination. The results showed statistically significant differences in AMD (*p* < 0.001), life satisfaction (*p* < 0.001), autonomy (*p* = 0.014), and relatedness (*p* = 0.001). Thus, in Figure 3, it is observed that adolescent NPs for both internet and mobile phone use presented higher scores in AMD (*p* = 0.009–0.035) and life satisfaction (*p* = 0.001–0.014) than those who presented OP or PU on the internet, mobile phones, or both. Furthermore, for measurements of AMD, adolescents with OP on internet use but NP on mobile phone use presented higher scores than those with PU on both the use of the internet and mobile phones (*p* = 0.047). It should also be noted that for autonomy (*p* = 0.003–0.023) and relatedness (0.004–0.018), adolescents with NP on internet use and PU on mobile phone use showed lower scores than the rest of the groups.

## 4. Discussion

The present research attempts to fill a gap in the previous scientific literature, in which it is unknown how the degree of problematic internet and mobile use affects the level of physical activity, kinanthropometric and body composition variables, AMD, psychological state, and physical fitness. In addition, it is not known whether the joint problematic use of the internet and the mobile phone has a more negative influence on these variables, nor whether there are differences according to the gender of the adolescents. For this reason, the aims of the present study were to determine the differences in the physical activity level, kinanthropometric and body composition variables, AMD, psychological state, and physical fitness according to gender and different levels of problematic use of the internet and mobile phones and to establish the differences in the physical activity level, kinanthropometric and body composition variables, AMD, psychological state, and physical fitness among adolescents when considering problematic use of the internet and mobile phones in combination.

Regarding the first objective, to determine the differences in the physical activity level, kinanthropometric and body composition variables, AMD, psychological state, and physical fitness according to gender and different levels of problematic use of the internet and mobile phones, the results found that females with internet or mobile phone PU had worse levels of physical activity, AMD, life satisfaction, and competence, as compared to OP and NP females; furthermore, with respect to the use of internet and mobile phone in males, the NPs had the highest score in life satisfaction and relatedness, although the PUs scored higher in handgrip and CMJ. The differences found in the psychological state in males and females are similar to the findings of previous research [11,15], but the most important results are the differences found in AMD and the level of physical activity of females, which could be explained by the different uses of the internet and mobile phones according to the adolescent’s gender, with playing video games being common among males, while among females the use of social networks is more common [52]. These social networks are full of healthy content related to exercise or nutrition; however, previous research has shown that knowing this information is not directly related to changing habits [53], in addition to the fact that users who abuse the use of social networks tend to show unhealthy behaviors, more prevalent in the case of females [54]. Furthermore, the information on healthy habits available on social networks is not scientifically proven, and adolescents are not able to look for information on healthy habits in reliable sources [55]. Therefore, although this information should be contrasted in future research, it could indicate the need to educate adolescents about the websites they could use to obtain relevant information related to healthy habits, especially for females.

The relevance of these results lies in the fact that no previous research has analyzed the differences in a specific nutritional pattern, such as AMD, as a function of the different degrees of problematic use of the internet and mobile phones. It is true that the relationship between AMD and kinanthropometric variables [21], physical condition [22], and physical activity [23] has been extensively studied, but this is a novel aspect that may be of great relevance for the healthy development of adolescents. Furthermore, the existence of gender differences, with females being more affected in terms of AMD when considering problematic internet and mobile phone use, establishes a line of research that should be confirmed in future studies.

It should be noted that differences were only found for males in handgrip and CMJ, with males with internet and mobile phone PU scoring higher. A possible explanation for these results lies in the fact that handgrip strength and CMJ are variables that measure muscle strength and power, and previous research has found an association between handgrip strength and aggressiveness only in males [56], with aggressiveness being more present in adolescents with problematic use of the internet and mobile phones [57,58]. Another possible explanation would be related to the sports practices of adolescents, as those who are more aggressive tend to participate in activities related to martial arts and team-based sports, where strength and power are mainly developed [59].

As for the second objective, to establish the differences in the physical activity level, kinanthropometric and body composition variables, AMD, psychological state, and physical fitness among adolescents when considering problematic use of the internet and mobile phones in combination, the results showed that adolescents with NP in both internet and mobile phone use scored higher in AMD and life satisfaction than those with OP or PU of the internet, mobile phone, or both. In those studies that conducted the analysis separately, the results were similar, with higher scores on nutritional habits and psychological state found for adolescents without problematic usage [6,11,14,15]. The results obtained are in line with the previous scientific literature and, although this is the first study that jointly analyzed the use of the internet and the mobile phone, it seems clear that a correct use of both technologies must be maintained if the aim is to achieve an adequate psychological development of adolescents, as well as an adequate acquisition of healthy habits that may be decisive in this and later stages of life. Therefore, this is a novel aspect of the present research, since no previous research had analyzed the joint use of both technologies and shows that it is necessary to teach adolescents the importance of the correct use of new technologies, since they seem to play a determining role in relation to fundamental variables for the correct development of adolescents.

However, one particularly relevant aspect that should be taken into consideration is that adolescents with NP in internet use, but PU of mobile phones showed lower scores in autonomy and relatedness as compared to the rest of the groups. Previous research has shown how integrated mobile phones have become in the adolescent population [4], and that mobile phone use from ages 8 to 17 years old is progressively increasing [60]. In this regard, it highlights the fact that there is an increase in the use of mobile phones at night, between the ages of 13 and 16, for texting, phoning, or messaging, which leads to serious sleep disturbances that affect mood, the possibility of depression, decreased self-esteem, and coping skills [61]; this may be the reason why adolescents are affected in their autonomy and relationships, as they are in a state of vulnerability that hinders their development in these areas. However, it should be noted that the sample size of the group of adolescents with NP with internet use, and PU of the mobile phone, is very small, which could be influencing the statistical analyses performed. Therefore, although future research is needed that attempts to establish the differences in the psychological variables of the adolescents by considering the possible problematic use of the internet and the mobile phone together, the main novelty of the present research shows the special relevance that the mobile phone could have in the differences found in the psychological state of the adolescent population.

The results obtained in the present investigation show that in females the problematic use of internet and mobile phone seems to have a greater impact on the level of physical activity, AMD, and psychological state, while in males, it influences their psychological state and physical condition. This is an important aspect to consider, since previous research has shown how the practice of physical activity is lower in adolescent females than in males [62], and problematic internet use may have a greater impact on the physical practice of females. In addition, it was not found that adolescents with problematic use or occasional problems of both mobile phones and the internet presented significant differences in any variable with respect to adolescents with problematic use or occasional problems with the internet or mobile phones exclusively.

Based on the results obtained, the first practical implication derived from the present investigation is that problematic use or occasional problems may influence the adolescents’ daily state of health, mainly affecting their level of physical activity or AMD, in the case of females, or psychological state in the case of both males and females. Furthermore, the second novel aspect lies in the fact that, although adolescents with NP on internet and mobile phone use showed higher scores in AMD and life satisfaction, the PU of the mobile phone could play a more determinant role in the psychological state of adolescents than internet use, as adolescents who showed NP on internet use, but PU of mobile phones obtained lower scores in autonomy and relatedness.

The present research is not free of limitations. As this is a descriptive study, it is not possible to verify the causal relationship between internet and mobile phone use and the changes in the variables analyzed. Although education centers were chosen from different areas and those with the largest samples of adolescent populations, the sampling was non-probabilistic by convenience. For the joint analysis of internet and mobile phone use, certain groups had a reduced sample size. The use of questionnaires, despite being validated, is always risky, as it depends on the subjective completion of the questionnaire by the adolescents. The adolescents (12–16 years old) in the present investigation are in the middle of puberty, so maturation may influence them. More specifically, in the case of females, those who are in the menstrual process and those who have not yet gone through this process have not been considered, which is a factor that could influence the results, as has been shown in previous studies in which the practice of physical activity was evaluated [63].

## 5. Conclusions

To conclude, problematic internet and mobile phone use could have different effects on adolescent males and females, with females being more affected in terms of physical activity level, AMD, and psychological state (life satisfaction and competence), while males are affected in terms of AMD, physical condition, and psychological state (life satisfaction and relatedness). Problematic internet use in females led to a lower level of physical activity, AMD, life satisfaction, and competence compared to NP and OP girls. In males, problematic internet use led to lower life satisfaction, but higher handgrip and CMJ scores. In terms of problematic mobile phone use, females showed worse AMD and life satisfaction, while males showed lower life satisfaction and relatedness, but higher CMJ scores. In addition, adolescents with NP with both internet and mobile phones showed higher AMD and life satisfaction scores than those who presented with OP or PU of the internet, mobile phone, or both. However, special attention should be paid in future research to adolescents with PU of mobile phones, as they seem to be more affected in their psychological state than the rest of the groups. The relevance of the results found lies in the fact that the use of mobile phones and the internet can have a negative effect on the physical activity, AMD, physical condition, and psychological state of adolescents, which are determining factors and should be considered in programs that seek to improve the health of this population from a global point of view. In addition, the effect sizes obtained in the present study are low for all the variables analyzed, so the results obtained should be taken with caution and show the need for future research in this area to delve deeper into the subject.

Based on the present investigation, future research should include objective measurement of the level of physical activity, since it would provide more information on the physical activity behavior of adolescents. Furthermore, the maturational process in which adolescents are immersed is a fundamental aspect to be considered due to the physical, psychological, and behavioral changes that occur during this stage, and future research should focus on the maturational stage.

## Figures and Tables

**Figure 1 healthcare-11-01706-f001:**
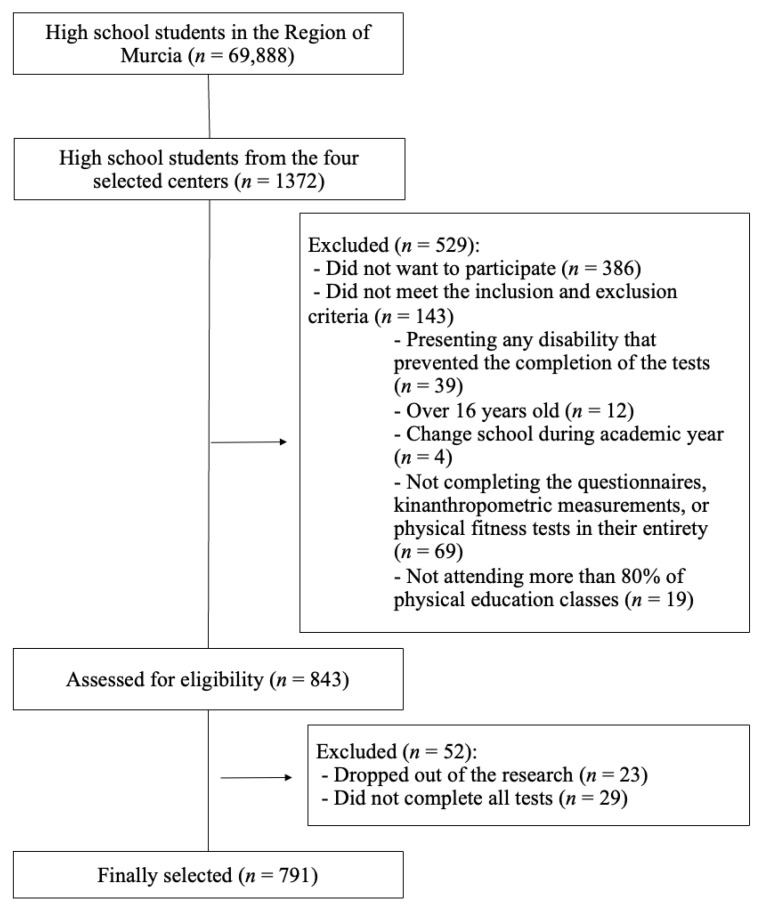
Sample selection flowchart.

**Figure 2 healthcare-11-01706-f002:**
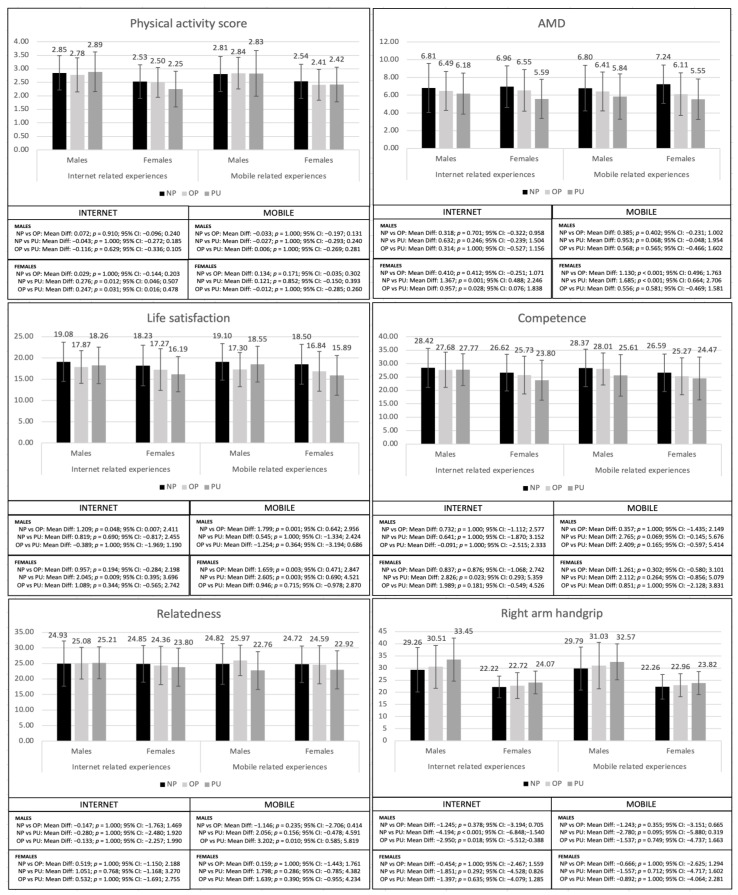
Bonferroni post-hoc analyses of the variables that showed significant differences between the different levels of problematic internet and mobile use in males and females.

**Figure 3 healthcare-11-01706-f003:**
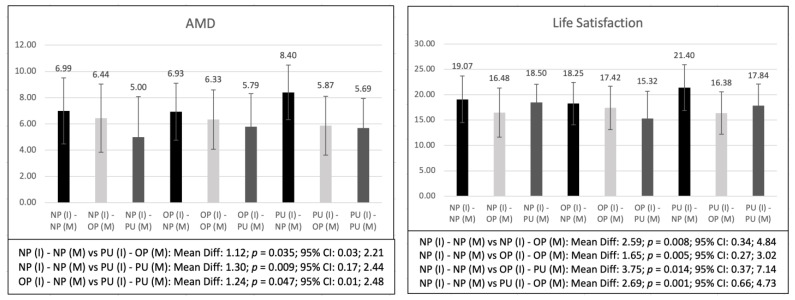
Bonferroni post-hoc analyses of the variables that showed significant differences when comparing adolescents with and without problematic use of both the internet and mobile phones.

**Table 1 healthcare-11-01706-t001:** Differences between males and females with different levels of problematic internet and mobile phone use in their physical activity level, kinanthropometric variables, AMD, psychological state, and physical condition.

		Internet Related Experiences (M ± SD)					Mobile Related Experiences (M ± SD)				
		NP (M:142) (F:150)	OP (M:181) (F:149)	PU (M:86) (F:83)	F, *p*	η^2^	NP (M:204) (F:164)	OP (M:144) (F:156)	PU (M:62) (F:61)	F, *p*	η^2^
Physical activity score	M	2.85 ± 0.64	2.78 ± 0.63	2.89 ± 0.73	1.000; *p* = 0.368	0.003	2.81 ± 0.65	2.84 ± 0.59	2.83 ± 0.84	0.122; *p* = 0.885	0.001
	F	2.53 ± 0.62	2.50 ± 0.55	2.25 ± 0.66	4.377; *p* = 0.013	0.012	2.54 ± 0.63	2.41 ± 0.57	2.42 ± 0.64	1.995; *p* = 0.137	0.005
Body mass (kg)	M	59.09 ± 15.52	60.54 ± 15.04	62.68 ± 12.19	1.744; *p* = 0.176	0.005	59.93 ± 14.77	60.85 ± 15.02	60.76 ± 14.62	0.243; *p* = 0.784	0.001
	F	53.56 ± 10.67	53.07 ± 9.81	55.43 ± 9.60	0.732; *p* = 0.481	0.002	52.70 ± 9.90	54.45 ± 10.36	54.53 ± 10.26	0.844; *p* = 0.430	0.002
Height (cm)	M	165.56 ± 9.99	166.96 ± 9.63	169.82 ± 8.39	5.884; *p* = 0.053	0.016	166.48 ± 9.91	167.16 ± 9.44	168.24 ± 9.23	0.851; *p* = 0.427	0.002
	F	159.33 ± 6.08	159.71 ± 6.48	160.34 ± 6.31	0.335; *p* = 0.716	0.001	158.98 ± 6.16	160.08 ± 6.34	160.76 ± 6.38	1.110; *p* = 0.330	0.003
BMI (kg/m^2^)	M	21.38 ± 4.78	21.60 ± 4.17	21.66 ± 3.26	0.167; *p* = 0.846	0.001	21.52 ± 4.19	21.62 ± 4.25	21.27 ± 3.59	0.120; *p* = 0.887	0.001
	F	21.13 ± 4.01	20.79 ± 3.14	21.58 ± 3.81	0.897; *p* = 0.408	0.002	20.85 ± 3.63	21.27 ± 3.66	21.09 ± 3.62	0.450; *p* = 0.638	0.001
Fat mass (%)	M	20.17 ± 11.46	20.27 ± 11.43	20.77 ± 10.68	0.083; *p* = 0.920	0.001	20.17 ± 10.78	20.74 ± 12.59	19.42 ± 8.71	0.306; *p* = 0.737	0.001
	F	25.81 ± 8.95	24.97 ± 7.47	26.25 ± 8.69	0.450; *p* = 0.638	0.001	25.22 ± 8.73	26.02 ± 7.79	24.83 ± 8.66	0.364; *p* = 0.695	0.001
Muscle mass (kg)	M	21.74 ± 5.08	22.08 ± 4.78	22.75 ± 4.14	1.407; *p* = 0.245	0.004	22.00 ± 5.06	22.13 ± 4.48	22.14 ± 4.60	0.056; *p* = 0.946	0.001
	F	15.54 ± 2.68	15.52 ± 2.98	16.12 ± 2.89	0.539; *p* = 0.583	0.001	15.33 ± 2.67	15.84 ± 2.96	16.03 ± 3.02	0.883; *p* = 0.414	0.002
Sum of three skinfolds (cm)	M	44.52 ± 25.98	44.63 ± 26.35	45.46 ± 23.49	0.035; *p* = 0.965	0.001	44.40 ± 24.25	45.64 ± 29.00	43.01 ± 19.94	0.219; *p* = 0.803	0.001
	F	60.00 ± 23.83	57.62 ± 20.56	61.21 ± 21.92	0.610; *p* = 0.544	0.002	58.64 ± 23.45	60.10 ± 20.77	57.93 ± 22.72	0.208; *p* = 0.812	0.001
Corrected arm girth (cm)	M	22.33 ± 3.13	22.60 ± 3.19	22.91 ± 2.58	1.044; *p* = 0.353	0.003	22.50 ± 3.24	22.63 ± 2.92	22.52 ± 2.83	0.109; *p* = 0.897	0.001
	F	19.99 ± 2.39	19.93 ± 2.18	20.25 ± 2.28	0.292; *p* = 0.747	0.001	19.85 ± 2.33	20.14 ± 2.22	20.15 ± 2.36	0.535; *p* = 0.586	0.001
Corrected thigh girth (cm)	M	41.82 ± 5.87	41.92 ± 4.79	42.39 ± 4.57	0.338; *p* = 0.713	0.001	41.94 ± 5.49	42.02 ± 4.84	41.86 ± 4.70	0.022; *p* = 0.979	0.001
	F	38.25 ± 3.96	38.11 ± 3.77	38.97 ± 3.91	0.769; *p* = 0.464	0.002	37.96 ± 3.95	38.55 ± 3.75	38.83 ± 4.03	0.917; *p* = 0.400	0.002
Corrected calf girth (cm)	M	30.13 ± 3.14	30.56 ± 2.91	30.58 ± 2.70	0.935; *p* = 0.393	0.003	30.45 ± 3.15	30.35 ± 2.77	30.36 ± 2.72	0.048; *p* = 0.953	0.001
	F	28.16 ± 3.09	28.17 ± 3.47	28.54 ± 2.41	0.371; *p* = 0.690	0.001	28.12 ± 3.11	28.28 ± 3.36	28.44 ± 2.46	0.211; *p* = 0.810	0.001
Waist girth (cm)	M	71.99 ± 9.80	72.67 ± 9.13	73.72 ± 7.75	0.945; *p* = 0.389	0.003	72.35 ± 9.47	73.08 ± 9.31	71.99 ± 6.91	0.434; *p* = 0.648	0.001
	F	66.39 ± 7.49	66.42 ± 6.68	67.89 ± 7.96	0.792; *p* = 0.453	0.002	65.91 ± 6.64	67.43 ± 7.79	66.58 ± 7.25	1.355; *p* = 0.259	0.004
Hip girth (cm)	M	89.58 ± 10.28	90.69 ± 10.38	92.72 ± 8.57	2.560; *p* = 0.078	0.007	90.07 ± 10.04	91.24 ± 10.42	91.09 ± 9.29	0.739; *p* = 0.478	0.002
	F	90.90 ± 8.60	90.47 ± 7.63	92.39 ± 7.71	0.941; *p* = 0.391	0.003	90.21 ± 7.94	91.58 ± 8.06	91.67 ± 8.56	1.023; *p* = 0.360	0.003
Waist-to-hip ratio	M	0.80 ± 0.04	0.80 ± 0.06	0.80 ± 0.04	0.745; *p* = 0.475	0.002	0.80 ± 0.06	0.80 ± 0.04	0.79 ± 0.03	1.208; *p* = 0.300	0.003
	F	0.73 ± 0.04	0.73 ± 0.04	0.73 ± 0.04	0.302; *p* = 0.739	0.001	0.74 ± 0.04	0.74 ± 0.04	0.73 ± 0.03	0.891; *p* = 0.411	0.002
AMD	M	6.81 ± 2.77	6.49 ± 2.19	6.18 ± 2.32	1.652; *p* = 0.192	0.004	6.80 ± 2.56	6.41 ± 2.19	5.84 ± 2.55	3.064; *p* = 0.047	0.008
	F	6.96 ± 2.35	6.55 ± 2.35	5.59 ± 2.21	6.978; *p* = 0.001	0.018	7.24 ± 2.16	6.11 ± 2.41	5.55 ± 2.29	13.114; *p* < 0.001	0.034
Life satisfaction	M	19.08 ± 4.63	17.87 ± 3.86	18.26 ± 4.32	2.939; *p* = 0.054	0.008	19.10 ± 4.30	17.30 ± 4.01	18.55 ± 4.20	7.004; *p* = 0.001	0.019
	F	18.23 ± 4.78	17.27 ± 4.89	16.19 ± 4.14	4.750; *p* = 0.009	0.013	18.50 ± 4.69	16.84 ± 4.68	15.89 ± 4.68	8.412; *p* < 0.001	0.022
Competence	M	28.42 ± 7.29	27.68 ± 6.55	27.77 ± 5.91	0.482; *p* = 0.618	0.001	28.37 ± 6.97	28.01 ± 5.96	25.61 ± 7.76	2.602; *p* = 0.075	0.007
	F	26.62 ± 6.85	25.73 ± 7.05	23.80 ± 7.42	3.590; *p* = 0.028	0.010	26.59 ± 6.98	25.27 ± 6.90	24.47 ± 7.97	2.245; *p* = 0.107	0.006
Autonomy	M	26.01 ± 7.44	26.18 ± 5.59	26.82 ± 4.99	0.368; *p* = 0.692	0.001	26.19 ± 6.37	26.68 ± 5.85	24.68 ± 6.84	1.537; *p* = 0.216	0.004
	F	25.28 ± 6.74	25.61 ± 6.02	23.61 ± 5.56	2.218; *p* = 0.110	0.006	25.53 ± 6.67	24.99 ± 5.87	24.11 ± 6.24	0.886; *p* = 0.413	0.002
Relatedness	M	24.93 ± 7.28	25.08 ± 5.13	25.21 ± 5.15	0.052; *p* = 0.950	0.001	24.82 ± 6.57	25.97 ± 4.86	22.76 ± 6.08	4.620; *p* = 0.010	0.012
	F	24.85 ± 5.91	24.36 ± 6.13	23.80 ± 6.11	0.691; *p* = 0.501	0.002	24.72 ± 5.88	24.59 ± 6.13	22.92 ± 6.14	1.449; *p* = 0.235	0.004
VO2 max. (ml/kg/min)	M	42.37 ± 5.48	42.10 ± 5.86	42.52 ± 5.14	0.210; *p* = 0.811	0.001	42.14 ± 5.66	42.13 ± 5.42	43.50 ± 5.92	1.253; *p* = 0.286	0.003
	F	36.96 ± 4.50	37.22 ± 4.06	35.78 ± 4.96	1.741; *p* = 0.176	0.005	37.25 ± 4.51	36.58 ± 4.18	36.50 ± 4.91	0.818; *p* = 0.442	0.002
Right arm handgrip (kg)	M	29.26 ± 9.13	30.51 ± 8.81	33.45 ± 8.86	7.193; *p* = 0.001	0.019	29.79 ± 8.87	31.03 ± 9.55	32.57 ± 7.39	2.871; *p* = 0.057	0.008
	F	22.22 ± 4.44	22.72 ± 5.32	24.07 ± 4.67	1.378; *p* = 0.253	0.004	22.26 ± 5.04	22.96 ± 4.74	23.82 ± 4.76	0.839; *p* = 0.433	0.002
Left arm handgrip (kg)	M	27.84 ± 8.58	28.27 ± 7.90	30.60 ± 7.79	4.004; *p* = 0.019	0.011	27.94 ± 8.24	28.80 ± 8.33	30.24 ± 6.99	2.250; *p* = 0.106	0.006
	F	20.79 ± 4.27	21.06 ± 4.31	21.48 ± 3.88	0.243; *p* = 0.785	0.001	20.65 ± 4.41	21.32 ± 3.97	21.30 ± 4.42	0.471; *p* = 0.625	0.001
Sit-and-reach (cm)	M	12.59 ± 6.34	12.51 ± 7.80	14.18 ± 8.44	1.043; *p* = 0.353	0.003	12.63 ± 6.94	12.70 ± 8.04	14.17 ± 8.39	0.596; *p* = 0.551	0.002
	F	18.90 ± 8.77	19.31 ± 8.84	19.70 ± 8.97	0.221; *p* = 0.802	0.001	19.16 ± 8.99	18.48 ± 8.69	22.38 ± 8.05	3.523; *p* = 0.055	0.009
CMJ (cm)	M	25.80 ± 7.38	26.09 ± 7.61	28.54 ± 6.93	4.310; *p* = 0.014	0.011	25.38 ± 8.02	27.28 ± 6.79	28.35 ± 5.96	5.795; *p* = 0.003	0.015
	F	20.16 ± 4.93	21.08 ± 5.11	21.35 ± 5.06	1.095; *p* = 0.335	0.003	20.23 ± 4.97	21.04 ± 5.04	21.66 ± 5.24	1.091; *p* = 0.336	0.003
20 m sprint (s)	M	3.70 ± 0.66	3.75 ± 0.49	3.67 ± 0.33	0.666; *p* = 0.514	0.002	3.71 ± 0.64	3.75 ± 0.41	3.67 ± 0.28	0.430; *p* = 0.651	0.001
	F	4.20 ± 0.40	4.12 ± 0.34	4.10 ± 0.65	1.293; *p* = 0.275	0.003	4.17 ± 0.40	4.17 ± 0.35	3.99 ± 0.75	2.350; *p* = 0.096	0.006

M: male; F: female; NP: no problems; OP: occasional problems; PU: problematic use; BMI: body mass index; AMD: adherence to Mediterranean diet; VO2 max: maximum oxygen consumption; CMJ: countermovement jump; η^2^: effect size.

**Table 2 healthcare-11-01706-t002:** Differences in the physical activity level, kinanthropometric variables, AMD, psychological state, and physical condition among adolescents with different levels of problematic internet and mobile phone use.

	NP (I)– NP (M) (*n* = 243)	NP (I)– OP(M) (*n* = 48)	NP (I)– PU (M) (*n* = 11)	OP (I)– NP (M) (*n* = 121)	OP (I)– OP (M) (*n* = 193)	OP (I)– PU (M) (*n* = 39)	PU (I)– NP (M) (*n* = 20)	PU (I)– OP (M) (*n* = 61)	PU (I)– PU (M) (*n* = 55)	*p*	η^2^
Physical activity score	2.69 ± 0.67	2.64 ± 0.54	2.56 ± 0.36	2.68 ± 0.62	2.63 ± 0.60	2.72 ± 0.66	2.82 ± 0.47	2.54 ± 0.74	2.60 ± 0.82	*p* = 0.864	0.005
Body mass (kg)	56.13 ± 12.93	57.44 ± 16.17	41.32 ± 6.75	57.64 ± 14.13	56.46 ± 12.21	59.99 ± 19.10	63.14 ± 10.54	60.37 ± 12.87	57.43 ± 9.90	*p* = 0.223	0.014
Height (cm)	162.45 ± 8.85	162.26 ± 8.37	152.55 ± 5.44	164.42 ± 9.81	163.19 ± 8.44	163.96 ± 11.05	166.48 ± 10.71	165.16 ± 9.72	165.12 ± 7.68	*p* = 0.106	0.018
BMI (kg/m^2^)	21.20 ± 4.15	21.66 ± 4.64	16.78 ± 3.08	21.19 ± 3.58	21.14 ± 3.75	21.98 ± 4.53	22.89 ± 3.70	22.02 ± 3.80	21.07 ± 3.14	*p* = 0.504	0.010
Fat mass (%)	22.65 ± 10.49	25.49 ± 10.97	16.84 ± 12.01	21.62 ± 9.45	22.43 ± 10.33	25.88 ± 11.03	30.45 ± 13.13	25.05 ± 11.12	21.02 ± 7.95	*p* = 0.061	0.020
Muscle mass (kg)	18.69 ± 5.19	17.91 ± 4.52	15.62 ± 0.01	19.70 ± 5.60	18.77 ± 4.83	19.04 ± 6.34	19.67 ± 4.21	19.77 ± 5.32	19.23 ± 4.47	*p* = 0.427	0.011
Sum of three skinfolds (cm)	51.47 ± 25.66	58.20 ± 27.42	40.00 ± 33.38	48.48 ± 22.89	50.80 ± 25.43	57.91 ± 27.57	68.87 ± 29.38	56.24 ± 26.06	48.28 ± 20.05	*p* = 0.133	0.017
Corrected arm girth (cm)	21.21 ± 3.11	20.81 ± 2.44	18.39 ± 0.03	21.54 ± 3.29	21.31 ± 2.95	21.24 ± 3.07	21.29 ± 2.35	21.76 ± 2.82	21.48 ± 2.79	*p* = 0.674	0.008
Corrected thigh girth (cm)	40.05 ± 5.42	39.79 ± 4.59	36.10 ± 1.82	40.47 ± 4.95	39.98 ± 4.53	40.55 ± 5.73	38.74 ± 3.83	41.15 ± 4.92	40.43 ± 4.22	*p* = 0.694	0.008
Corrected calf girth (cm)	29.26 ± 3.41	28.51 ± 2.38	26.66 ± 1.30	29.78 ± 3.17	29.39 ± 3.53	28.55 ± 3.22	28.42 ± 3.59	29.50 ± 2.89	29.79 ± 2.52	*p* = 0.285	0.013
Waist girth (cm)	69.00 ± 8.68	69.89 ± 11.15	62.77 ± 7.96	70.19 ± 9.33	69.48 ± 8.07	70.40 ± 9.85	76.16 ± 7.86	72.01 ± 9.47	69.14 ± 6.59	*p* = 0.228	0.014
Hip girth (cm)	90.05 ± 9.17	91.75 ± 10.65	79.50 ± 9.05	90.17 ± 9.23	90.57 ± 8.79	92.35 ± 12.28	93.31 ± 7.84	93.48 ± 8.91	91.47 ± 7.21	*p* = 0.136	0.017
Waist-to-hip ratio	0.77 ± 0.05	0.76 ± 0.06	0.79 ± 0.01	0.78 ± 0.07	0.77 ± 0.05	0.76 ± 0.05	0.82 ± 0.06	0.77 ± 0.05	0.76 ± 0.05	*p* = 0.133	0.017
AMD	6.99 ± 2.51	6.44 ± 2.59	5.00 ± 3.07	6.93 ± 2.17	6.33 ± 2.26	5.79 ± 2.51	8.40 ± 2.07	5.87 ± 2.24	5.69 ± 2.24	*p* < 0.001	0.041
Life satisfaction	19.07 ± 4.60	16.48 ± 4.84	18.50 ± 3.54	18.25 ± 4.19	17.42 ± 4.30	15.32 ± 5.38	21.40 ± 4.51	16.38 ± 4.19	17.84 ± 4.25	*p* < 0.001	0.053
Competence	27.75 ± 7.07	26.71 ± 6.95	25.50 ± 2.12	27.24 ± 7.04	26.88 ± 6.45	23.74 ± 8.62	27.20 ± 4.03	25.69 ± 6.78	25.87 ± 7.41	*p* = 0.057	0.023
Autonomy	25.93 ± 7.05	24.85 ± 6.49	8.50 ± 2.12	25.80 ± 5.40	26.17 ± 5.99	24.42 ± 6.20	26.40 ± 3.29	25.43 ± 5.18	24.96 ± 6.02	*p* = 0.014	0.026
Relatedness	25.00 ± 6.62	24.98 ± 5.78	9.00 ± 2.83	24.35 ± 5.58	25.35 ± 5.48	21.05 ± 5.72	24.20 ± 4.09	25.05 ± 5.90	23.96 ± 5.53	*p* = 0.001	0.034
VO2 max. (ml/kg/min)	39.82 ± 5.78	38.26 ± 4.93	41.97 ± 8.86	40.22 ± 5.65	39.81 ± 5.57	39.33 ± 6.65	40.51 ± 5.38	38.30 ± 5.71	40.16 ± 6.43	*p* = 0.369	0.012
Right arm handgrip (kg)	25.99 ± 7.98	24.09 ± 7.54	18.90 ± 1.98	27.27 ± 8.53	26.79 ± 8.29	27.19 ± 8.94	28.00 ± 16.06	28.96 ± 9.15	28.87 ± 7.01	*p* = 0.056	0.023
Left arm handgrip (kg)	24.36 ± 7.74	23.71 ± 6.78	17.45 ± 1.06	25.29 ± 7.44	24.83 ± 7.44	25.17 ± 7.98	26.76 ± 12.62	25.98 ± 7.86	26.28 ± 7.12	*p* = 0.423	0.011
Sit-and-reach (cm)	15.96 ± 8.49	15.36 ± 8.24	13.25 ± 0.35	14.73 ± 8.70	15.98 ± 9.06	16.66 ± 9.21	14.00 ± 7.39	15.17 ± 8.77	19.02 ± 9.26	*p* = 0.243	0.014
CMJ (cm)	23.03 ± 6.98	22.24 ± 6.21	21.90 ± 5.37	23.20 ± 7.93	24.35 ± 6.34	23.37 ± 7.44	23.44 ± 7.81	24.58 ± 7.76	25.68 ± 6.18	*p* = 0.143	0.016
20 m sprint (s)	3.94 ± 0.63	4.06 ± 0.42	4.09 ± 0.16	3.86 ± 0.52	3.94 ± 0.42	4.02 ± 0.45	4.09 ± 0.53	3.98 ± 0.47	3.76 ± 0.62	*p* = 0.138	0.016

I: internet related experiences; M: mobile phone related experiences; NP: no problems; OP: occasional problems; PU: problematic use; BMI: body mass index; AMD: adherence to Mediterranean diet; VO2 max: maximum oxygen consumption; CMJ: countermovement jump; η2: effect size.

## Data Availability

The data presented in this study are available on request from the corresponding author. The data are not publicly available because they contain information that could compromise the privacy of research participants, but they are available from the corresponding author upon reasonable request.

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
