# Peer review of "The Degree of Problematic Technology Use Negatively Affects Physical Activity Level, Adherence to Mediterranean Diet and Psychological State of Adolescents"

_healthcare, 2023, doi:10.3390/healthcare11121706_

Round 1

Reviewer 1 Report

Congratulating the authors for this work, I leave here some of my questions, hoping to contribute to its improvement.

- I think that having two perfectly defined objectives, hypotheses would not be necessary.

- These ages certainly included girls before and after menarche, which, if controlled, could have led to slightly different results, or allowed other justifications for the results obtained.

- Line 126 - the inclusion criteria that appear here, should be before figure 1, as this would facilitate its understanding.

- In figure 1, in the third box, referring to "Excluded", the reasons for considering that the students "did not meet the inclusion and exclusion criteria" should be stated.

- Conclusions with only 10 lines is too small for an article with 18 pages and a lot of data.

- Finally, the role of the Mediterranean diet is not clear, given that it appears in the title, but then practically does not exist in the study.

Author Response

Congratulating the authors for this work, I leave here some of my questions, hoping to contribute to its improvement.

+ Dear reviewer, thank you very much for reviewing our manuscript and making these comments for improvement. We have addressed all of them and believe that the manuscript has been substantially improved.

I think that having two perfectly defined objectives, hypotheses would not be necessary.

+ Thank you very much for your input. The hypotheses have been removed from the introduction and discussion.

These ages certainly included girls before and after menarche, which, if controlled, could have led to slightly different results, or allowed other justifications for the results obtained.

+ Thank you very much for your great contribution. We fully agree with your indication, and it was something that was not taken into account, therefore, it has been included as a limitation of the present study.

Line 126 - the inclusion criteria that appear here, should be before figure 1, as this would facilitate its understanding.

+ Thank you for your input. The paragraph including inclusion and exclusion criteria has been moved before Figure 1.

In figure 1, in the third box, referring to "Excluded", the reasons for considering that the students "did not meet the inclusion and exclusion criteria" should be stated.

+ Thank you very much for your input. The Figure 1 has been modified.

Conclusions with only 10 lines is too small for an article with 18 pages and a lot of data.

+ Thank you very much for your contribution. It is true that the conclusions were brief. All relevant information has been included and a practical application has been included that may be useful for professionals in the sports, educational and health fields.

Finally, the role of the Mediterranean diet is not clear, given that it appears in the title, but then practically does not exist in the study.

+ Thank you very much for your contribution. As stated in the introduction, no previous research on AMD and problematic internet and mobile phone use has been found. However, a paragraph has been included in the discussion addressing this variable and may be of relevance to the field.

Again, thank you very much for the time spent on this review. We have addressed all requested changes and the manuscript has improved considerably. If any other consideration is necessary, please let us know.

Reviewer 2 Report

General comments:

The aims of the study were to analyze the differences in the physical activity level, kinanthropometry and body composition variables, adherence to Mediterranean diet, psychological state, and physical fitness of adolescents with different levels of problematic use of the internet and mobile phones according to gender; and to establish differences in these variables when considering problematic internet and mobile use together. The authors found that adolescent males and females with problematic internet and/or mobile phone use presented a worse psychological state, and females had a lower level of physical activity and adherence to Mediterranean diet.

Overall, the manuscript is clear, relevant for the field and written in a well-structured manner.

Introduction:

1. Could authors define what "problematic use" of mobile phones and internet is in this context before mentioning what its rate is? (Line 32).

2. "According to kinanthropometric and derived variables, adolescents with more than three hours ..." (lines 50-52) seems incomplete. Please check. 

3. Is an adherence to the Mediterranean diet (AMD) [lines 60-62] the only indicator of bad nutritional habits among this age group? Could the authors provide a justification for choosing this over other dietary patterns?

Author Response

General comments:

The aims of the study were to analyze the differences in the physical activity level, kinanthropometry and body composition variables, adherence to Mediterranean diet, psychological state, and physical fitness of adolescents with different levels of problematic use of the internet and mobile phones according to gender; and to establish differences in these variables when considering problematic internet and mobile use together. The authors found that adolescent males and females with problematic internet and/or mobile phone use presented a worse psychological state, and females had a lower level of physical activity and adherence to Mediterranean diet.

Overall, the manuscript is clear, relevant for the field and written in a well-structured manner.

+ Dear reviewer, thank you very much for taking the time to review the manuscript and for your comments. We have attended to all of them with the aim of substantially improving it.

Introduction:

  1. Could authors define what "problematic use" of mobile phones and internet is in this context before mentioning what its rate is? (Line 32).

+ Thank you very much for your contribution. We have included information about problematic internet and mobile use.

  1. "According to kinanthropometric and derived variables, adolescents with more than three hours ..." (lines 50-52) seems incomplete. Please check. 

+ Thank you very much, the information in this sentence has been completed.

  1. Is an adherence to the Mediterranean diet (AMD) [lines 60-62] the only indicator of bad nutritional habits among this age group? Could the authors provide a justification for choosing this over other dietary patterns?

+ Thank you very much for your contribution. It has been indicated in the introduction that this is the most used pattern in adolescents, together with the questionnaire par excellence (KIDMED), since it is related to numerous healthy habits in this population.

Dear reviewer, thank you for reviewing our manuscript and helping us to improve it substantially. We have addressed all your suggestions. If any other suggestions are needed, please let us know.

Reviewer 3 Report

Thank you for your submitted the systematic review entitled, The Degree of Problematic Technology Use Negatively Affects Physical Activity Level, Adherence to Mediterranean Diet, and Psychological State of Adolescents.  

ABSTRACT

·       The objectives of the study has not been sufficiently defined.

·       Clarify the subjects’ level and background

·       Why not use average of age, height, body mass and body mass index of participants

·       Could be a relevant conclusion of the present study to find what is important to know.

INTRODUCTION

·       The development of the introduction needs to be more hypotheses driven and develop the questions leading up to the section in the methods section.

·       The Authors should clarify the actual heritage of this manuscript. I am concerned about the originality of the article

METHOD

·       How was sample size determined? (Sampling technique!)

·       It is important that you help the reader with the context of the study concerning the subjects’ background, and information that will allow other investigators to put your data into context with the literature.

·       What about the inclusion and exclusion criteria?

RESULTS

·       The results are clearly described.

DISCUSSION

·       The discussion needs to reflect what you found, how it relates to the literature. Make sure the paper’s importance and need is clear to the reader.

CONCLUSION

  • Finally, improve the conclusion section: it is important to suggest possible future studies.

Author Response

Thank you for your submitted the systematic review entitled, The Degree of Problematic Technology Use Negatively Affects Physical Activity Level, Adherence to Mediterranean Diet, and Psychological State of Adolescents.  

+ Dear reviewer, thank you for reviewing our manuscript.

ABSTRACT

The objectives of the study has not been sufficiently defined.

+ Thank you for your input. The aims have been specified in full.

Clarify the subjects’ level and background.

+ Thank you very much. The information has been included.

Why not use average of age, height, body mass and body mass index of participants.

+ Thank you for your comment. This information has been included.

Could be a relevant conclusion of the present study to find what is important to know.

+ Thank you very much. The general conclusion of the study has been included.

INTRODUCTION

The development of the introduction needs to be more hypotheses driven and develop the questions leading up to the section in the methods section.

+ Thank you very much for your contribution. Information on aspects that have been left unanalysed in previous research has been included throughout the introduction.

The Authors should clarify the actual heritage of this manuscript. I am concerned about the originality of the article.

+ Thank you very much for your great contribution. A paragraph has been included prior to the aims reporting the literature gap and the novel aspects to be contributed by the present investigation.

METHOD

How was sample size determined? (Sampling technique!)

+ Thank you very much. This information has been included in the design and participants section.

It is important that you help the reader with the context of the study concerning the subjects’ background, and information that will allow other investigators to put your data into context with the literature.

+ Thank you for your great contribution. We have contextualized the research by adding information in the participants section.

What about the inclusion and exclusion criteria?

+ Thank you for your input. The inclusion and exclusion criteria have been repositioned so that they are more visible to the reader.

RESULTS

The results are clearly described.

+ Thank you very much.

DISCUSSION

The discussion needs to reflect what you found, how it relates to the literature. Make sure the paper’s importance and need is clear to the reader.

+ Thank you very much for your contribution. Gaps in the scientific literature and what this research contributes in this regard have been indicated.

CONCLUSION

Finally, improve the conclusion section: it is important to suggest possible future studies.

+ Thank you very much for your great contribution. Future possible investigations have been included.

Again, thank you for your interest in the manuscript and for providing this revision that has helped us to improve it substantially. Any other modifications that may be necessary, we are at your disposal.

Round 2

Reviewer 3 Report

Thank you so much for your feedback. I really appreciate the details you shared in the manuscript